# NGFI-A Binding Protein 2 Promotes EGF-Dependent HNSCC Cell Invasion

**DOI:** 10.3390/cancers11030315

**Published:** 2019-03-06

**Authors:** Jinkyung Kim, Sung-Min Kang, Su Young Oh, Heon-Jin Lee, Inhan Lee, Jae-Chan Hwang, Su-Hyung Hong

**Affiliations:** 1Department of Microbiology and Immunology, School of Dentistry, Kyungpook National University, Daegu 700-412, Korea; kimjk0925@knu.ac.kr (J.K.); dkdkdk43@knu.ac.kr (S.-M.K.); oohsuy@knu.ac.kr (S.Y.O.); heonlee@knu.ac.kr (H.-J.L.); 2Research Division, MIRCORE, Ann Arbor, MI 48105, USA; inhanlee99@gmail.com (I.L.); hwangjc@umich.edu (J.-C.H.)

**Keywords:** head and neck squamous cell carcinoma, epidermal growth factor, NGFI-A binding protein 2, early growth response 1, Sp1, the cancer genome atlas

## Abstract

NGFI-A binding protein 2 (NAB2) represses the transcriptional activation of early growth response protein-1 (EGR1), a tumor-suppressor. However, Epidermal Growth Factor (EGF) promotes tumor progression even with significant EGR1 upregulation. The molecular mechanism through which NAB2 is involved in cancer is largely unknown. Therefore, we evaluated how the NAB2-mediated suppression of EGR1 facilitates head and neck squamous cell carcinoma (HNSCC) cancer progression, in association with Sp1, which competes with EGR1 as a transcriptional regulator. The effect of NAB2 on EGR1/SP1 binding to the consensus promoter sequences of *MMP2* and *MMP9* was evaluated by chromatin immunoprecipitation (ChIP) and promoter luciferase assay. The correlation between *EGR1*-*NAB2* expression and metastatic status was investigated using The Cancer Genome Atlas (TCGA) for HNSCC patients. Our data showed that NAB2 knockdown in FaDu and YD-10B HNSCC cells alleviated EGF-dependent increase of Matrigel invasion. In addition, NAB2 upregulation in EGF-treated FaDu cell diminishes EGR1 transcriptional activity, resulting in the upregulation of Sp1-dependent tumor-promoting genes. TCGA data analysis of 483 HNSCC tumors showed that higher levels of both *EGR1* and *NAB2* mRNA were significantly associated with metastasis, corresponding to in vitro results. Our data suggest that NAB2 upregulation facilitates EGF-mediated cancer cell invasion through the transactivation of Sp1-dependent tumor-promoting genes. These results provide insight into the paradoxical roles of EGF-EGR1 in cancer progression.

## 1. Introduction

Despite the many advances in cancer treatment, the 5-year survival rate for patients with head and neck squamous cell carcinoma (HNSCC) has improved only marginally. Further, cancer metastasis, rather than primary tumors, is responsible for most of cancer-related deaths [1]. Therefore, characterizing the signaling cascades that are responsible for tumor invasion and metastasis in HNSCC will help to identify aggressive tumors early in the disease process.

Early growth response 1 (EGR1), a zinc finger transcription factor, is rapidly and transiently induced by a diverse set of extracellular stresses. Analysis of certain human tumor cells and tissues indicate that Egr1 exhibits a prominent tumor suppressor function [2,3,4,5]. Further, sustained EGR1 expression can result in the inhibition of tumor cell invasion and tumor growth in in vitro and in vivo analysis [6]. It also downregulates the expression of genes involved in tumor metastasis, including matrix metalloproteinase 2 (*MMP2*) [7]. Interestingly, Krones-Herzig showed a uniformly accelerated development of skin tumors in EGR1-null mice. However, there is little data showing that EGR1 directly inhibits cancer progression in vivo.

EGFR plays a key role in integrating the signaling events that lead to the regulation of cell growth [8,9]. Its expression has been correlated with poor prognosis for HNSCC patients [10]. Epidermal Growth Factor (EGF) stimulates HNSCC migration and invasion through various molecular mechanisms [11,12]. Due to its critical roles in cell survival and proliferation, EGFR is a target of anticancer treatments [13]. Unfortunately, EGFR tyrosine kinase inhibitors have yielded limited results for patients with HNSCC [14,15]. Interestingly, EGF is known to promote cancer progression through EGR1 upregulation [16]. In contrast, another study showed that EGF-dependent EGR1 upregulation inhibits the proliferation of lymphoma and lung cancer cells [17,18]. However, these paradoxical roles of EGF in cancer progression have still not been evaluated. Therefore, we first sought to uncover how EGR1 permits tumor progression in EGF-treated HNSCC cells.

The transcription factor specificity protein 1 (Sp1) is known to have an important function in cell growth control. The target genes of this protein are key factors in oncogenesis [19], cell proliferation [20], and tumorigenesis [21]. Kubosaki et al. showed that Sp1 binding to the promoter region of target genes was dramatically changed near EGR1 binding sites [22]. Furthermore, a previous study reported that EGR1 binding competes with Sp1 binding because of their similar consensus sequences [23]. These data suggested interplay between EGR1 and Sp1, which results in multiple responses with respect to EGR1 downstream genes. NGFI-A binding protein 2 (NAB2), an inducible modulator of transcription, represses the transcriptional activation mediated by EGR1 [24]. The human *NAB2* gene is localized to chromosome 12q13.3–14.1 [24], a region that is rearranged in several tumor tissues. However, there have been no studies demonstrating EGR1-NAB2 regulatory mechanisms associated with Sp1 during cancer progression.

Our previous study showed that in NHSCC cells, oxytocin, which sustained EGR1 expression, preferentially inhibits tumor cell invasion in an EGFR-dependent manner [25]. However, as shown in this study, EGF was found to significantly upregulate EGR1 expression and increase cancer cell invasion. Previous studies showed that EGR1 downregulates *MMP2* [7] and *MMP9* [17] by binding directly to their promoter in cancer cells. Interestingly, Sp1 is also known to upregulate MMP2 and MMP9 in cancer cells [26,27]. In the present study, we investigated whether EGF-mediated NAB2 upregulation attenuates the EGR1-dependent transcriptional inhibition of *MMP2* and *MMP9*, resulting in increased expression of MMPs via Sp1 transcriptional activation.

## 2. Results

### 2.1. EGF Upregulates EGR1 Expression in HNSCC Cells

The effect of EGF on the expression of EGR1 in FaDU and YD-10B cells was evaluated by qPCR and/or Western blot analysis. In FaDU cells, EGR1 mRNA and protein expression increased remarkably within 30 min of EGF treatment, then reverted to original levels after 24 h (Figure 1A). As expected, EGF downregulated the expression of *E-cadherin* mRNA within 3 h (Figure 1B). A previous study showed that EGF increases the invasion of ovarian cancer cells through EGR1 upregulation [28]. We thus investigated the effect of exogenous EGR1 overexpression on FaDU cell invasion. As shown in Figure 1C, FaDU cells overexpressing EGR1 showed decreased Matrigel invasion as compared to that in control cells. Similar results in terms of EGF-dependent EGR1 upregulation and the effect of EGR1 on Matrigel invasion were observed with YD-10 B cells (Figure 1D).

### 2.2. NAB2 Knockdown in HNSCC Cells Alleviates EGF-Dependent Increase of Matrigel Invasion

We then analyzed the mRNA and protein expression of NAB2, a co-repressor of EGR1, under the same experimental conditions. EGF was found to upregulate the mRNA and protein expression of NAB2 within 1 h (Figure 2A). To evaluate the effect of NAB2 on EGF-dependent cell invasion, cells transfected with a specific siRNA mixture targeting NAB2 were pretreated with EGF (Figure 2B). Interestingly, siNAB2 transfection for 48 h significantly decreased EGF-dependent Matrigel invasion in FaDU and YD-10B cells (* *p* < 0.01 and ** *p* < 0.005, respectively; Figure 2C,D).

Next, we evaluated the effect of NAB2 on Matrigel invasion using FaDU spheroids. Figure 3A shows the downregulation of *NAB2* mRNA in FaDU spheroids after siNAB2 transfection for 10 days in 96-well plates. Figure 3B shows representative images of Matrigel invasion under the same conditions. After 10 days, the spheroids treated with siNAB2 transfection showed a significant decrease in Matrigel invasion in response to EGF (* *p* < 0.05; Figure 3C).

### 2.3. NAB2 Knockdown Relieves the EGF-Dependent Upregulation of MMP2 and MMP9

As shown in Figure 4A, EGF treatment was found to upregulate the mRNA and protein expression of MMP2 and MMP9 in FaDU cells. However, siNAB2 pretreatment diminished the EGF-dependent upregulation of both MMPs. An earlier study showed that NAB2 modulates the expression of EGR1-target genes via a direct interaction with EGR1 [29]. Therefore, we used ChIP-qPCR to evaluate if NAB2 affects the competitive binding of Sp1 and EGR1 to the promoters of target genes. Figure 4B shows a schematic representation of the promoter regions containing consensus EGR1/Sp1-binding sites (black arrowheads), which were detected using the AliBaba 2.1 program. As shown in Figure 4C, Sp1 binding to the consensus sequences of the *MMP2* and *MMP9* promoters was decreased in NAB2-knockdown FaDU cells. In contrast, EGR1 binding to the same promoter sequences was increased under the same conditions (Figure 4D). Furthermore, transient overexpression of EGR1 decreased Sp1 binding to the consensus sequences of *MMP2* and *MMP9* promoters. However, co-overexpression of EGR1 and NAB2 recovered Sp1 binding to the promoter regions (Figure 4E). To further evaluate the effect of EGR1, NAB2, and Sp1 on *MMP2* and *MMP9* promoter activity, we performed promoter luciferase assays after modulating the expression of these three genes. As shown in Figure 4F, *MMP2* and *MMP9* promoter activity decreased with EGR1 overexpression but was recovered with NAB2 overexpression. However, Sp1 knockdown with the co-overexpression of EGR1 and NAB2 decreased the activities of these promoters, supporting the ChIP data.

### 2.4. Molecular Signatures Reflected by the TCGA Dataset of HNSCC Patient Tumor Tissues

To infer the correlation of EGR1-NAB2 expression and HNSCC metastasis, we performed unsupervised hierarchical clustering analysis based on the mRNA expression of these markers, together with *EGFR* expression as a reference. Whereas EGR1 can be upregulated by EGF, it was found to be more closely related to NAB2 than EGFR (Figure 5A). As shown in our in vitro experiment, multivariate analysis identified a cluster of overall high EGR1 and low NAB2 expression (blue box in Figure 5A; total of 121 samples) and overall high EGR1 and high NAB2 expression (red box in Figure 5A; total of 94 samples). Both clusters had moderate EGFR expression showing that EGFR status is not a determining factor for these two clusters. As shown in Figure 5B, the unsupervised hierarchical clustering of EGFR, EGR1, and NAB2 expression values for all 483 patients showed significantly fewer metastatic tumors (53 metastatic/121 total tumors; hypergeometric *p* = 0.018) in the high EGR1/low NAB cluster, whereas the high EGR1/high NAB2 cluster was associated with significantly more metastatic tumors (53 metastatic/94 total tumors; hypergeometric *p* = 0.044), with the total cluster number being six (purple line in Figure 5A showing the tree cut). Human papilloma virus (HPV) is a potential causative agent of HNSCC. Therefore, we included HPV infection status for the 248 samples as previously identified [30] in Figure 5A, and showed that subgroups did not depend on HPV status.

Without the EGFR reference (EGR1 and NAB2 expression correlations), we identified the existence of similar subgroups, such as 81 tumors in a high EGR1/low NAB2 expression cluster (blue box in Figure 5C) and 31 tumors in a high EGR1/high NAB2 cluster (red box in Figure 5C), with the total cluster number being six (the tree cut indicated by the purple line in Figure 5C). Whereas the high EGR1/low NAB2 cluster presented with significantly fewer metastatic tumors (32 metastatic/81 total tumors; hypergeometric *p* = 0.005), the high EGR1/high NAB2 cluster was associated with a significant increase in metastatic tumors (20 metastatic/31 total tumors; hypergeometric *p* = 0.043; Figure 5D). Whereas the number of samples in each cluster decreased as compared to EGFR-EGR1-NAB2 clustering (Figure 5B), the statistical significance increased after determining metastatic status by performing EGR1-NAB2 clustering (Figure 5D). In comparison, this type of metastatic signature was not identified by clustering analysis based on EGFR-EGR1 expression (no significance in the high EGR1 /high EGFR group, red box in Appendix A), indicating that subgroups newly identified by EGR1-NAB2 expression signatures are clinically important.

From these data, we can infer that EGF-dependent co-upregulation of EGR1 and NAB2 represses the transcriptional regulatory activity of EGR1, resulting in an increase in Sp1-dependent *MMP* gene expression (Figure 6B). In contrast, EGR1 upregulation alone, via EGF, might inhibit MMP expression, resulting in decreased cell invasion (Figure 6A).

## 3. Discussion

An earlier study showed that a mutation resulting in a constitutively active type of EGFR upregulates EGR1 [31]. However, mutant EGFR is oncogenic in non-small cell lung cancer [32]. Although previous studies have shown the paradoxical effect of EGR1 upregulated by EGF on cancer progression [16,17,18], none have reported on the molecular mechanisms that explain this phenomenon. Furthermore, our previous study showed that oxytocin inhibits HNSCC cell invasion in an EGFR-EGR1-dependent manner [25]. In the present study, EGF treatment led to the upregulation of EGR1 expression as well as an increase in HNSCC cell invasion, with the remarkable downregulation of E-cadherin. Therefore, we aimed to evaluate how EGF increases cancer cell invasion during conditions of EGR1 upregulation. One of the crucial differences between the actions of oxytocin and EGF is that only EGF was found to upregulate NAB2, resulting in the inhibition of EGR1-dependent transcriptional regulation. Despite the potential importance of NAB2 in modulating EGR1 activity, its regulation remains only partly understood. Furthermore, even though the importance of EGR1 is recognized with respect to cancer progression, the detailed mechanism regarding the action of NAB2 in association with EGR1 is largely unknown.

Smalley et al. suggested that a 3-D spheroid model resembles human tumors in vivo by establishing a gradient of oxygen and nutrients [33]. Therefore, the 3-D Matrigel invasion assay provides several advantages over the current in vitro invasion assays, such as mimicking the tumor microenvironment or micro-metastasis, as well as the high reproducibility of spheroid size [34]. We used the 3-D Matrigel invasion method in a 96-well plate to evaluate the effect of EGR1/NAB2 on HNSCC cell invasion upon EGF treatment. Interestingly, NAB2 knockdown significantly diminished the EGF-dependent increase in 3-D spheroid invasion. Interestingly, transient NAB2 knockdown in spheroids was achieved for more than 10 days (Figure 3A).

The cell invasion process involves the degradation of the basement membrane by proteinases, which typically include MMP2 and MMP9. Because MMP2 and MMP9 are associated with the metastatic potential of HNSCC, they represent attractive targets for therapy. An earlier study showed that EGR1 directly inhibits *MMP2* in breast cancer cells by binding to its promoter [35]. Further, EGF-dependent EGR1 upregulation inhibits the growth of thymic lymphoma by suppressing MMP9 production from stromal cells [17]. Therefore, EGR1 has the potential to inhibit cell invasion via the downregulation of MMP2 and MMP9. A previous study showed that EGR1 binding competes with Sp1 due to similar consensus sequences in the promoters of target genes [23]. Moreover, an earlier study showed that Sp1 upregulates MMP2 and MMP9 in cancer cells [26,27]. It is therefore necessary to understand in detail the molecular mechanism(s) through which EGR1-NAB2 and Sp1 regulate the transcription of *MMP2* and *MMP9*. We evaluated whether EGF-dependent NAB2 upregulation diminishes the transcriptional regulatory effect of EGR1, thereby upregulating Sp1-target gene expression. HNSCC cells treated with EGF upregulated EGR1 and NAB2 expression, which are presumed to form a complex that inhibits EGR1 binding to DNA consensus sequences. This process might result in the replacement of EGR1 with Sp1, which binds to consensus promoter sequences. Several findings in the present study, based on ChIP assays with an EGR1 or Sp1 antibody, as well as *MMP* promoter luciferase assays, support this assumption. There is no doubt that the inhibition of MMP would reduce cancer metastasis. In this respect, the phenomenon that NAB2 is involved in promoting the expression of MMP by EGF is very interesting. Further study would be needed for the kinetic analysis of MMP by NAB2 expression in an EGF-dependent manner.

Regarding these data, we evaluated whether the mRNA expression pattern of *EGR1*-*NAB2* is associated with metastasis in HNSCC patients. The diverse nature of cancer progression and metastasis makes it difficult to translate in vitro studies into in vivo experiments. TCGA provides multiple types of datasets from a large patient cohort, facilitating the assessment of molecular mechanisms based on patient data. Based on TCGA data analysis of HNSCC patient tissues, a subgroup comprising a large population presenting with higher EGR1 expression and lower NAB2 levels was significantly associated with decreased metastasis. In contrast, a subgroup with high mRNA expression of both *EGR1* and *NAB2* comprised more metastatic patients (Figure 5D). This signature was strong enough to be apparent based on the analysis of all samples. Further, there were 71 samples with EGR1 values higher than the median values of all samples and NAB2 values higher than 3rd quartile values. These samples were significantly associated with increased metastasis (42 metastatic vs 29 non-metastatic), with a hypergeometric distribution *p*-value of 0.032, showing that increased NAB2 expression abolishes the protective role of EGR1. Since cancer involves the dysregulation of multiple molecular pathways, the categorization of clinically important subgroups is crucial. The clustering of subgroups based on EGR1-NAB2 expression correlation values may or may not be clinically important. However, we found that EGR1-NAB2 subgroups were associated with metastatic signatures, supporting our in vitro data, which suggests dual functional mechanisms for EGF in metastasis.

## 4. Materials and Methods

### 4.1. Chemicals and Reagents

DMEM, FBS, and penicillin/streptomycin were obtained from Gibco (Invitrogen, Carlsbad, CA, USA). SYBR green PCR master mix was obtained from Takara Biotechnology (Dalian, Japan). Rabbit anti-EGR1 antibody (Cell Signaling Technology Cat# 4154, RRID:AB_2097035) was purchased from Cell Signaling (Danvers, MA, USA). Rabbit anti-E-cadherin antibody (Abcam Cat# 1702-1, RRID:AB_562059) was acquired from Abcam (Cambridge, UK). Rabbit anti-NAB2 (Santa Cruz Biotechnology Cat# sc-22815, RRID:AB_2298032), anti-Sp1 (Santa Cruz Biotechnology Cat# sc-59, RRID:AB_2171050), HRP-conjugated anti-β-actin antibody (Santa Cruz Biotechnology Cat# sc-47778 HRP, RRID:AB_2714189), and secondary antibodies were acquired from Santa Cruz Biotechnology (Santa Cruz, CA, USA). Rabbit anti-MMP9 (Abcam Cat# 2551-1, RRID:AB_1267245) and mouse anti-MMP2 (Abcam Cat# ab7032, RRID:AB_2145819) antibodies were purchased from Abcam (Cambridge, MA, USA). Primary and secondary antibodies were diluted 1:1000 and 1:5000, respectively. All other reagents were obtained from standard commercial sources.

### 4.2. Cell Culture

FaDU (ATCC Cat# HTB-43, RRID:CVCL_1218) and YD-10B (KCLB Cat# 60503, RRID:CVCL_8929) human oral cancer cell lines were obtained from the American Type Culture Collection (Manassas, VA, USA) and Korean Cell Line Bank (Seoul, Korea), respectively. They were maintained in Dulbecco’s Modified Eagle Medium (DMEM) containing 10% Fetal Bovine Serum and 1% penicillin/streptomycin solution. Cells were grown at 37 °C in a 5% CO_2_ humidified atmosphere. The cell lines were tested for contamination every two months with CellSafe Mycoplasma PCR detection kit (Cat. No. CS-D, Cellsafe Co., Yongin, South Korea).

### 4.3. Transfection of siRNA or Overexpression Vectors

Cells were suspended in serum-free medium, which was followed by transfection with control siRNA (sc-37007) or an siNAB2 mixture (sc-36014) consisting of three specific targeting oligonucleotides (Santa Cruz Biotechnology) using an electroporation system (Bio-Rad Laboratories, Inc., Hercules, CA, USA). For exogenous EGR1 overexpression, the pCDNA3.1-EGR1 vector was used [36]. The pcMV6-AC-GFP NAB2 vector (Origin Technologies, Rockville, MD, USA) was used for NAB2 overexpression.

### 4.4. Real-Time PCR (qRT-PCR)

Total RNA was extracted using QIAzol, followed by reverse-transcription to cDNA using a first-strand cDNA synthesis kit (Applied Biosystems, Foster City, CA, USA). The following primers were used for qPCR: EGR1 forward, 5′-CAGGAGTGATGAACGCAAGA, reverse, 5′-GGGATGGGTAGGAAGAGAGG; NAB2, forward, 5′-CACATCCCTGCTAAAGCTGAA, reverse, 5′-GTCGAAACGGCCATAGATGAT; E-cadherin forward, 5′-CGACCCAACCCAAGAATCTA, reverse, 5′-CTCCAAGAATCCCCAGAATG; MMP2 forward, 5′-AGCTGCAACCTGTTTGTGCTG, reverse, 5′-CGCATGGTCTCGATGGTATTCT; MMP9 forward, 5′-ACGACGTCTTCCAGTACCGAGA, reverse, 5′-TAGGTCACGTAGCCCACTTGGT; GAPDH forward, 5′-AGATCATCAGCAATGCCTCCTG reverse, 5′-CTGGGCAGGGCTTATTCCTTTTCT. qPCR was carried out using an ABI 7500 real-time PCR system (Applied Biosystems). Calculations were performed based on the values of the Δcycle threshold (ΔCt), which was determined by normalizing the average Ct value of each treatment to that of the endogenous *GAPDH* control and then calculating the 2^−ΔΔCt^ value for each treatment.

### 4.5. Western Blot Analysis

Cells were lysed with 200 µL of PRO-PREP protein extraction solution (iNtRON Biotechnology, Seoul, South Korea). Protein concentrations were estimated using Coomassie protein assay reagent (Thermo Scientific, Rockford, IL, USA). Then, protein samples were electrophoresed on Sodium dodecyl sulphate-polyacrylamide gel, which was followed by transfer to nitrocellulose membranes. The membranes were blocked with 5% skim milk in Tris-buffered Saline, and subsequently incubated overnight with primary antibody at 4 °C. HRP-conjugated secondary antibody was applied for 1 h and then the blot was washed in TBST (TBS and 0.1% Tween 20). Proteins of interest were detected using Enhanced chemiluminescence (ECL) detection reagents. The relative intensities of the bands were analyzed using ImageJ software (https://imagej.nih.gov/ij/download.html).

### 4.6. Matrigel Invasion Assays

Control siRNA- or specific siRNA-transfected FaDU cells were added to Matrigel-coated transwells. After culturing for 48 h, the cells were washed twice with PBS and stained with 0.2% crystal violet. Cells that had migrated to the lower surface of the filters were counted. The invasion index was calculated as the fold-change in the number of invaded cells in the experimental groups compared to that in the control group.

Invasion assays using Matrigel-embedded 3-D spheroids performed in a 96-well plate with one spheroid per well is a highly reproducible, standardized method [37,38]. Briefly, FaDU cells were seeded in a 96-well U-bottom ultra-low attachment plate (6000 cells per well) (Corning, NY, USA) and cultured for 4 days to form spheroids (>700 μm in diameter). Lipofectamine 2000 and the siNAB2 mixture were added to the spheroids cultured in a 96-well plate. After 8 h, the spheroids were centrifuged and 50 μL of Matrigel matrix was added directly to each well containing 100 μL medium to provide a semi-solid matrix into which the tumor cells could invade from the spheroid body. Matrigel invasion was monitored over a period of 10 days by phase-contrast microscopy (5× magnification) and then quantified by measuring the average length of the tube-like structures extending from the surface of each spheroid.

### 4.7. Chromatin Immunoprecipitation (ChIP) Analysis

We also investigated whether EGF-dependent NAB2 upregulation results in altered transcriptional regulatory efficiency for Sp1 and EGR1. For this, we performed ChIP analysis with an Sp1 or EGR1 antibody using *MMP2* and *MMP9* genes in FaDU cells with control siRNA or siNAB2 pretreatment. In addition, Sp1 ChIP analysis was performed in EGR1- and/or NAB2-overexpressing cells. After 48 h, cross-linking was performed by formaldehyde. Cell pellets were resuspended in SDS lysis buffer and sonicated for chromatin fragmentation to an average length of less than 500 bp. ChIP was performed overnight at 4 °C with anti-Sp1/anti-EGR1 antibody or normal rabbit IgG. Next, 50 μL of protein A agarose/salmon sperm DNA beads (Merck Millipore, Billerica, MA, USA) was added and incubated for 4 h. Reverse crosslinking was carried out by incubating the samples overnight with 0.3 M NaCl at 65 °C. RNA and unbound protein were removed by adding RNase A and proteinase K, respectively. DNA was extracted using the PCR purification kit (Qiagen, Valencia, CA, USA) and suspended in 50 μL of Tris-ethylenediaminetetraacetic acid buffer. qPCR was performed using the immunoprecipitated chromatin or input chromatin with primers flanking the EGR1 binding sequences of each promoter (MMP2-1, −4377 to −4363; MMP2-2, −1489 to −1484; MMP2-3, −411 to −406; MMP9-1, −6106 to −6097; MMP9-2, −1746 to −1737).

### 4.8. Promoter-Luciferase Assays

The MMP2-1 and MMP9-1 promoter fragments were synthesized from human genomic DNA (TAKARA BIO Inc., Shiga, Japan) The amplified PCR products were cloned into the pGL4.70 vector (Promega Corporation, Fitchburg, WI, USA,), upstream of the *Renilla* luciferase coding sequence. For the reporter assay, pGL4.70 with either *MMP2* or *MMP9* and the pcDNA3.1 control vector or pcDNA3.1-EGR1 overexpression plasmids were co-transfected via electroporation. The pGL3-basic firefly (*Photinus pyralis*) luciferase plasmid was co-transfected as a normalization control. The MMP2-1 and MMP9-1 mutant promoter constructs were synthesized using the combined overlap extension PCR method [39]. All assays were performed using the dual luciferase assay (Promega) according to the manufacturer’s protocol.

### 4.9. TCGA RNAseq Data Analysis

We downloaded all available clinical supplement data for TCGA-HNSCC project participants from the NCI’s GDC (National Cancer Institute’s Genomic Data Commons) Data Portal. Using the clinical supplement data and in-house software, we classified the patients as either metastatic or non-metastatic based on their AJCC (American Joint Committee on Cancer) clinical N score, where a score of N0 is considered non-metastatic and a score of N1–N3 is considered metastatic. Patients with a clinical N score of NX were removed from our study. Based on our classification of metastatic and non-metastatic HNSCC patients, we downloaded FPKM-UQ (Fragments Per Kilobase of transcript per Million mapped reads upper quartile) normalized RNA-seq data from tumors of the two groups for expression value comparison across all samples. Log_2_ transformed FPKM-UQ data were used to perform unsupervised hierarchical clustering using the R package heatmap (https://CRAN.R-project.org/package=pheatmap), the clustering distance was measured based on Pearson correlations.

### 4.10. Statistical Analysis

Significant variation analysis was used to carry out parametric two-tailed non-paired *t*-tests. All analyses were performed using Origin 8.0 (OriginLab, Northampton, MA, USA) and *p*-values ≤0.05 were considered statistically significant.

## 5. Conclusions

Our data suggest that upregulation of NAB2, a corepressor of EGR1, permits EGF-mediated cancer invasion during conditions of EGR1 overexpression through the transactivation of Sp1-dependent tumor-promoting genes. Our data might provide insight into understanding the paradoxical roles of EGF-EGR1 in cancer progression.

## Figures and Tables

**Figure 1 cancers-11-00315-f001:**
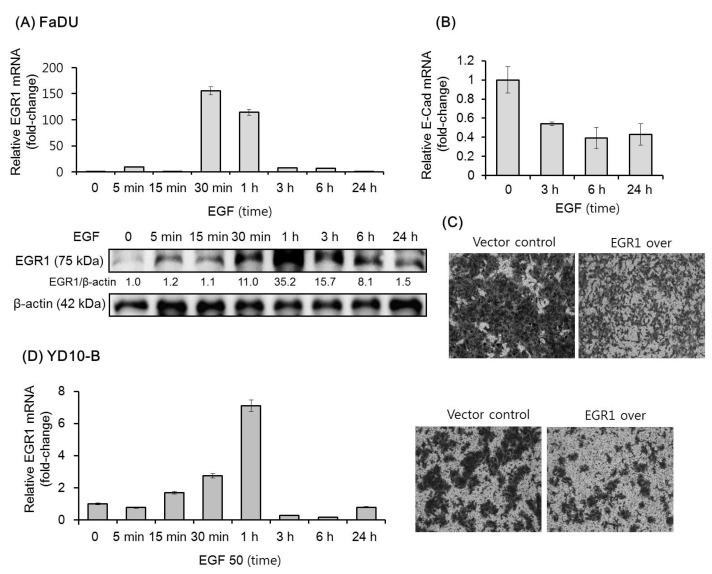
Effect of Epidermal Growth Factor (EGF) on the expression of EGR1 in head and neck squamous cell carcinoma (HNSCC) cells. (**A**) Time-dependent effect of EGF (50 ng/mL) administration over 24 h on the expression of EGR1 mRNA and protein as assessed by qPCR and Western blot analysis. (**B**) *E-cadherin* mRNA levels were analyzed under the same conditions. (**C**) Transwell Matrigel invasion assay was performed to assess the effect of EGR1 overexpression (over) on FaDU cell invasion. EGR1 overexpression vector- or control vector-transfected FaDU cells were added to each Matrigel-coated transwell. After culturing for 48 h, the cells were stained with 0.2% crystal violet in 10% ethanol, and cell invasion was monitored by phase-contrast microscopy (5× magnification). (**D**) EGF-dependent *EGR1* mRNA upregulation and the effect of EGR1 overexpression on Transwell Matrigel invasion using YD-10B cells were evaluated using the same conditions (5× magnification). The results shown are from two or three independent experiments, with each bar representing the standard deviation. Please add magnification for Figure 1 C and D.

**Figure 2 cancers-11-00315-f002:**
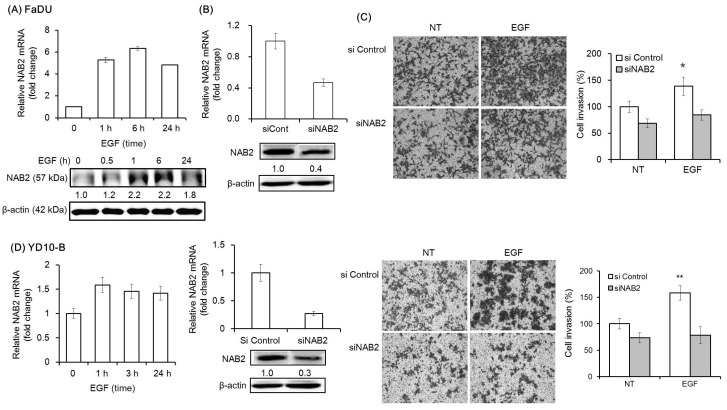
Effect of NAB2 on EGF-dependent head and neck squamous cell carcinoma (HNSCC) cell invasion. (**A**) NAB2 mRNA and protein expression following EGF treatment (50 ng/mL) for 24 h was analyzed in FaDU cells by qPCR and Western blot analysis. (**B**) FaDU cells were transfected with an siNAB2 mixture (40 nm/mL) for 48 h, after which the mRNA and protein expression of NAB2 were analyzed. (**C**) Two-dimensional Transwell Matrigel invasion assays were performed to analyze the effect of NAB2 knockdown, followed by EGF treatment, on FaDU cell invasion. Control siRNA- or siNAB2-transfected FaDU cells were added to each Matrigel-coated Transwell. After culturing for 48 h, the cells were washed twice with phosphate-bufferd saline and stained with 0.2% crystal violet in 10% ethanol (5× magnification). The cell invasion index was calculated as the difference between the number of invading cells in the siNAB2-transfected group compared to that in the control siRNA-transfected group (* *p* < 0.01). (**D**) Matrigel invasion assays using YD-10B cells were performed with the same conditions (5× magnification) (** *p* < 0.005). The results shown are from two or three independent experiments, with each bar representing the standard deviation.

**Figure 3 cancers-11-00315-f003:**
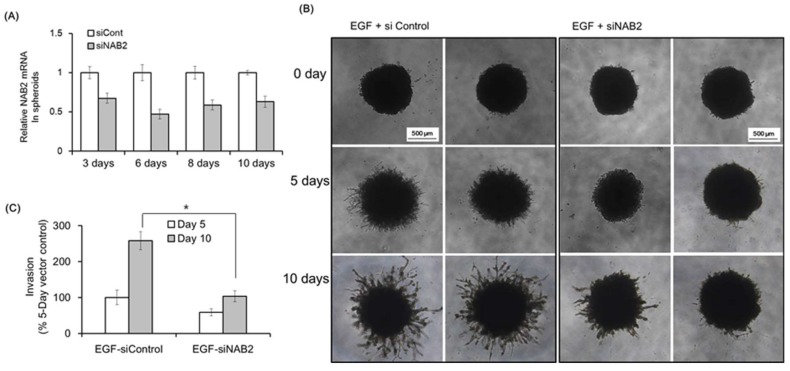
Effect of NAB2 on EGF-dependent FaDU spheroid invasion. (**A**) FaDU cells were seeded in a 96-well ultralow attachment U-bottom plate (6000 cells per well) and cultured for 4 days to form one spheroid per well (>700 μm in diameter). Control siRNA or siNAB2 was transfected into FaDU spheroids using Lipofectamine 2000 for 10 days and *NAB2* mRNA expression was evaluated by qPCR. (**B**) The Matrigel invasion of FaDU spheroids transfected with siNAB2 was investigated after EGF treatment (50 ng/mL). FaDU spheroids cultured for 5 days were transfected with control siRNA or siNAB2 using Lipofectamine 2000. After 8 h, the spheroids were centrifuged and 50 μL of Matrigel matrix was added directly to each well (100 μL of medium) to provide a semi-solid matrix into which tumor cells could migrate from the spheroid body. Matrigel invasion was monitored over a period of 10 days by phase-contrast microscopy (5× magnification). (**C**) Matrigel invasion was quantified based on the average length of tube-like structures protruding from the surface of each spheroid (* *p* < 0.005). The results shown are from two or three independent experiments, with each bar representing the standard deviation.

**Figure 4 cancers-11-00315-f004:**
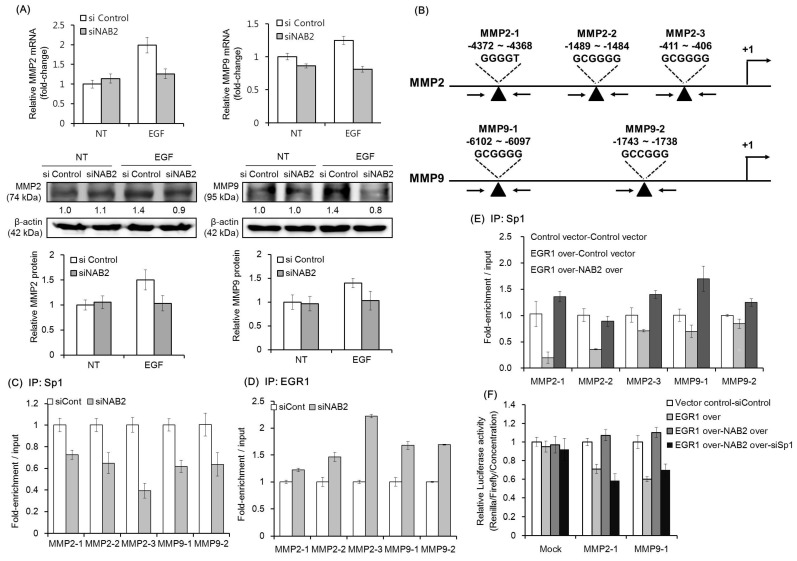
Effect of knockdown or overexpression of EGR1, NAB2, and Sp1 on MMP expression. (**A**) FaDU cells were transfected with control siRNA or siNAB2 for 24 h, and then subjected to EGF treatment (50 ng/mL) for another 24 h. Transfected cells were analyzed for mRNA and protein expression of MMP2 and MMP9. Protein expression was quantified and plotted in three independent experiments. (**B**) A schematic representation of the promoter regions containing the consensus EGR1/Sp1-binding sites (black arrowheads). FaDU cells were transfected with control siRNA or siNAB2 and chromatin was immunoprecipitated with (**C**) anti-Sp1 antibody or (**D**) anti-EGR1 antibody. The resulting immunoprecipitates were analyzed by qPCR to detect the consensus promoter sequences. (**E**) *EGR1*/*NAB2* genes were overexpressed in FaDU cells, after which chromatin was immunoprecipitated with an anti-Sp1 antibody. The resulting immunoprecipitates were analyzed by qPCR to detect the consensus promoter sequences. (**F**) For luciferase reporter assays using MMP2-1 and MMP9-1, promoter fragments were synthesized and cloned into the pGL4.70 vector. After 24 h of vector transfection, dual luciferase assays were performed to determine the effect of different combinations of EGR1/NBA2 overexpression (over) and/or Sp1 knockdown. The results shown are based on two or three independent experiments, with each bar representing the standard deviation.

**Figure 5 cancers-11-00315-f005:**
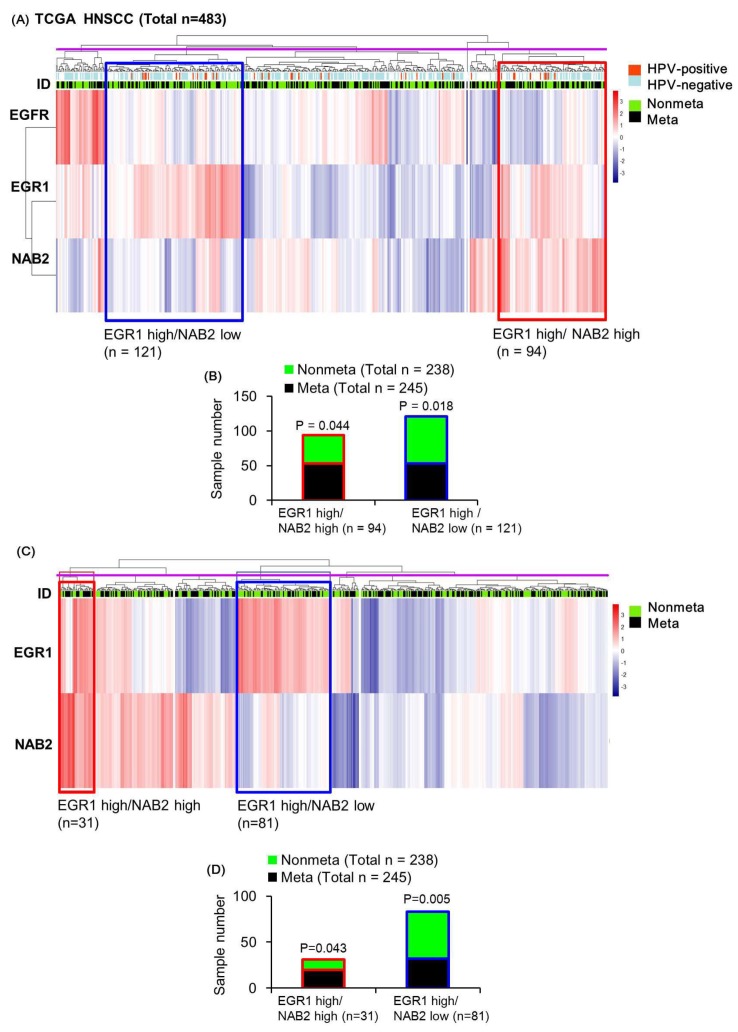
Heatmap representation of the relative mRNA expression levels of *EGFR*/*EGR1*/*NAB2* in the The Cancer Genome Atlas (TCGA) population of head and neck squamous cell carcinoma (HNSCC) specimens. Unsupervised hierarchical clustering with distance measured based on Pearson correlations was performed using log_2_-transformed FPKM-UQ normalized mRNA expression data for *EGFR*, *EGR1*, and *NAB2* for all patients. Tumors from metastatic and non-metastatic patients were annotated with green and black bars, respectively. (**A**) The subgroup indicated by a red box was found to display high *EGR1* mRNA expression and high *NAB2* mRNA expression. In contrast, the subgroup indicated by the blue box represents high *EGR1* mRNA and low *NAB2* mRNA expression. (**B**) The distribution of meta-nonmeta patients according to *EGF*/*EGR1*/*NAB2* expression in the heatmap is shown as a bar graph. (**C**) Heatmap representation of the relative mRNA expression levels of *EGR1*/*NAB2* according to human papilloma virus (HPV) infection in the TCGA population. The subgroup indicated by a red box was found to have high *EGR1* mRNA expression and high *NAB2* mRNA expression. In contrast, the blue box shows a subgroup with high *EGR1* mRNA and low *NAB2* mRNA expression. (**D**) The distribution of meta-nonmeta patients according to *EGR1*/*NAB2* expression in the heatmap is shown as a bar graph.

**Figure 6 cancers-11-00315-f006:**
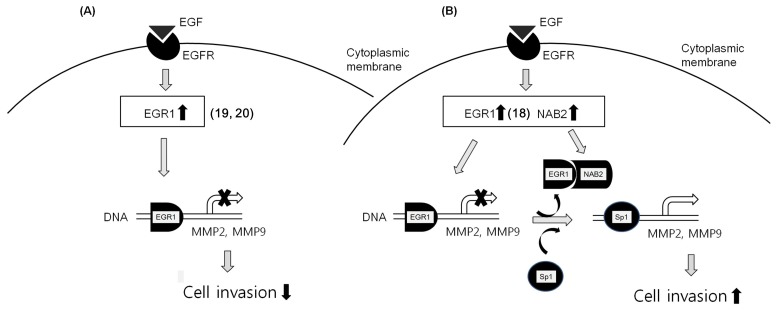
Putative schematic of EGF-EGR1-NAB2-dependent cancer cell invasion. (**A**) EGF-dependent EGR1 upregulation inhibits cancer progression by downregulating MMP. (**B**) However, EGF-dependent co-upregulation of EGR1 and NAB2 is presumed to form a complex in some cancer cells. This might result in the prevention of EGR1 binding to DNA consensus sequences, and its replacement with Sp1. Therefore, NAB2, a co-repressor of EGR1, permits EGF-mediated cancer invasion even with EGR1 upregulation.

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
