# Peer review of "NGFI-A Binding Protein 2 Promotes EGF-Dependent HNSCC Cell Invasion"

_cancers, 2019, doi:10.3390/cancers11030315_

Round 1
Reviewer 1 Report
The authors examine the relationship of ERG1 and NAB2 in EGF-mediated HNSCC metastasis. In particular, the authors are examining the paradox that ERG1 upregulation causes a decrease in proliferation of some cancer cells, but it is overexpressed in many HNSCCs. Based on some solid biochemical analysis, the authors build a strong case for the relationship of these two proteins. There findings will generally be of interest to cancer biologists.
Strengths of the manuscript include the use of multiple cells lines, complimentary knockdown and overexpression strategies, and multiple approaches to assess cell biology. Importantly, the authors examine the model they generate in human tissues. However, there are some components of the paper that diminish the overall enthusiasm.
Major concern:
1. The heading for Section 2.2 states: EGF-dependent NAB2 upregulation increases Matrigel invasion. The experiment shows there is a EGF-dependent NAB2 upregulation. However, the authors knockdown NAB2 and decrease Matrigel invasion. The wording needs to reflect the experiment.
2. In section, the authors argue that EGF-dependent upregulation of MMP2 and MMP9 is reversed by NAB2. The data would be strengthened by a kinetic analysis of the effect of NAB2 knockdown.
3. In Figure 4A, the immunoblot does not convincingly show differences in MMP2 protein levels. There is quantification of proteins levels from the single experiment. The addition of a graph showing the average and standard deviation of the protein levels would strengthen the findings.
4. The bottom left and bottom middle panels of figure 4 are unclear. Since there is no letter marking the panels. The panels should be lettered and a description should be in the Figure legends.
5. In Figure 5A – the writing below the red box needs editing.
Author Response
Comments and Suggestions for Authors
Reviewer 1
The authors examine the relationship of ERG1 and NAB2 in EGF-mediated HNSCC metastasis. In particular, the authors are examining the paradox that ERG1 upregulation causes a decrease in proliferation of some cancer cells, but it is overexpressed in many HNSCCs. Based on some solid biochemical analysis, the authors build a strong case for the relationship of these two proteins. There findings will generally be of interest to cancer biologists.
Strengths of the manuscript include the use of multiple cells lines, complimentary knockdown and overexpression strategies, and multiple approaches to assess cell biology. Importantly, the authors examine the model they generate in human tissues. However, there are some components of the paper that diminish the overall enthusiasm.
Major concern:
1. The heading for Section 2.2 states: EGF-dependent NAB2 upregulation increases Matrigel invasion. The experiment shows there is an EGF-dependent NAB2 upregulation. However, the authors knockdown NAB2 and decrease Matrigel invasion. The wording needs to reflect the experiment.
Response: Thank you for the comment. We revised this heading as follows:
2.2. NAB2 knockdown in HNSCC cells alleviates EGF-dependent increase of Matrigel invasion
2. In section, the authors argue that EGF-dependent upregulation of MMP2 and MMP9 is reversed by NAB2. The data would be strengthened by a kinetic analysis of the effect of NAB2 knockdown.
Response: Thank you for critical comment. As is well known, MMP plays an important role in the progression of multiple cancers including HNSCC. Therefore, inhibition of MMP is expected to reduce cancer metastasis. In this respect, the phenomenon that NAB2 is involved in promoting the expression of MMP by EGF is very interesting, and we plan to further investigate the kinetic analysis of MMP by NAB2 expression in an EGF-dependent manner. We are establishing a stable cell lines with FaDu which expressing EGR1-NAB2 for this study, but it takes more time than we thought. Therefore, in this study, we would like to focus more on the study of obtaining hints about the relationship between EGF-EGR1-NAB2. This point is stated in the discussion section of the text.
3. In Figure 4A, the immunoblot does not convincingly show differences in MMP2 protein levels. There is quantification of proteins levels from the single experiment. The addition of a graph showing the average and standard deviation of the protein levels would strengthen the findings.
Response: We performed this experiment in three independent times. We added a protein expression graph with the averages and standard deviation values in Fig. 4A as suggested. This is also mentioned in the legend of this figure.
4. The bottom left and bottom middle panels of figure 4 are unclear. Since there is no letter marking the panels. The panels should be lettered and a description should be in the Figure legends.
Response: The panel you pointed out is the electrophoresis gel image of the PCR product after immunoprecipitation. It is not an essential image and we decided to delete this panel because there is not enough space due to adding protein quantification graph in the (A) panel of this figure 4.
5. In Figure 5A – the writing below the red box needs editing.
Response: We regret our mistake. Thank you for the kind comment.
Reviewer 2 Report
This is an investigation of how EGF1 expression results in invasion of HNC cells. Authors show that NAB2 is important as it upregulates MMPs and that if either is knocked down, the invasion effect is suppressed.
This is a very well done manuscript. Flows nicely with appropriate controls and replicates. Only minor comment is that FaDu are known to be an abnormal HNSCC cell line (FA patient) and I see they do use a second cell line. Could be stronger if primary HNSCC cells were used or if ex vivo or mouse experiments were done. With that said, they do relate to TCGA data which is a nice addition.
Author Response
Comments and Suggestions for Authors
Reviewer 2
This is an investigation of how EGF1 expression results in invasion of HNC cells. Authors show that NAB2 is important as it upregulates MMPs and that if either is knocked down, the invasion effect is suppressed.
This is a very well done manuscript. Flows nicely with appropriate controls and replicates. Only minor comment is that FaDu are known to be an abnormal HNSCC cell line (FA patient) and I see they do use a second cell line. Could be stronger if primary HNSCC cells were used or if ex vivo or mouse experiments were done. With that said, they do relate to TCGA data which is a nice addition.
Response: Thank you very much for your evaluation of this paper. As you pointed out, FaDu is a tumor cell line derived from a hypopharyngeal site of FA patient. Nevertheless, there is no doubt that FaDu is one of the most used cell lines in HNSCC studies due to its malignant characteristics. In the present study, we tested several HNSCC cell lines for three-dimensional spheroid formation. Of these, FaDu exhibited the most reproducible results, which is why we proceeded this study with FaDu. Furthermore, several studies to establish new HNSCC cell lines used FaDu as one of the references HNSCC cell lines.
As you suggested, we are preparing to identify the results of this manuscript through the primary cell culture in consideration of the cancer microenvironment. In addition, we are planning to make stable FaDU cell lines in which NAB2 and/or EGR1 gene expression has been manipulated for in vivo mouse experiment. We sincerely hope that the results of this study would be well confirmed by further study and that we will be able to publish subsequent article. Recently, TCGA data analysis has been widely used in cancer research and gives more information than we expected. Thank you again for understanding its importance.
Reviewer 3 Report
The MS by Kim et al proposes a mechanisms whereby the up regulation fo NAB2 can help explain EGF-mediated cell invasion in HNSCC. Overall, the data are well presented and the experiments are clear. However, there is a tendency to overstate the importance to the data.
The text is not always easy to follow. For example, in the abstract, “The correlation between EGR1-NAB2 expression and metastatic status was determined using The Cancer Genome Atlas (TCGA) for HNSCC patients. NAB2 knockdown with EGF reduced cell invasion. “ This does not make a lot of sense and it would be good to clarify more exactly what the authors intend to say. They should be precise and say exactly what the data show. If it is matrigel invasion in vitro using one cell line they mean, this should be clear.
Overall there is a need to be more exact.
Introduction:
There is not a lot of published evidence for EGR1 inducing p53. Most of the sited refs provide circumstantial evidence. There are, for example, no animal models cited to support the claim of EGR1 in cancer development. It is preferable for the reader to get an objective view of what is known about EGR1. This does not by any means make the reported data less important.
Results:
Explain the rational for the cell lines used in this study.
I do not agree with the following statement. “To reveal the effect of EGR1 and NAB2 expression patterns on HNSCC metastasis, we performed unsupervised hierarchical clustering analysis based on the mRNA expression of these markers, together with EGFR expression as a reference. “ This is not correct. Data bases support or suggest a role in metastasis. Experimental approaches are needed to reveal the role.
Author Response
Comments and Suggestions for Authors
Reviewer 3
The MS by Kim et al proposes a mechanisms whereby the up regulation for NAB2 can help explain EGF-mediated cell invasion in HNSCC. Overall, the data are well presented and the experiments are clear. However, there is a tendency to overstate the importance to the data.
The text is not always easy to follow. For example, in the abstract, “The correlation between EGR1-NAB2 expression and metastatic status was determined using The Cancer Genome Atlas (TCGA) for HNSCC patients. NAB2 knockdown with EGF reduced cell invasion. “ This does not make a lot of sense and it would be good to clarify more exactly what the authors intend to say. They should be precise and say exactly what the data show. If it is matrigel invasion in vitro using one cell line they mean, this should be clear.
Response: Thank you for your kind comment. The abstract has been revised more clearly.
Introduction:
There is not a lot of published evidence for EGR1 inducing p53. Most of the sited refs provide circumstantial evidence. There are, for example, no animal models cited to support the claim of EGR1 in cancer development. It is preferable for the reader to get an objective view of what is known about EGR1. This does not by any means make the reported data less important.
Response: Thank you for your kind attention. We fully understand your concerns. We have reviewed some references for the effect of EGR1 on cancer progression more clearly with red color in the introduction part.
Results:
Explain the rational for the cell lines used in this study.
Response: FaDu, one of the most popular cell lines for studying HNSCC is derived from a pharyngeal site. It is highly malignant and suitable for performing the study for cancer progression including three-dimensional spheroid experiments. We used a second cell line, YD-10B which has been derived from tongue cancer tissues of patients. We expect these two cell lines to be able to represent head and neck cancer, including the oral cavity.
I do not agree with the following statement. “To reveal the effect of EGR1 and NAB2 expression patterns on HNSCC metastasis, we performed unsupervised hierarchical clustering analysis based on the mRNA expression of these markers, together with EGFR expression as a reference. “ This is not correct. Data bases support or suggest a role in metastasis. Experimental approaches are needed to reveal the role.
Response: Thank you for the comment. We revised this sentence as follow.
“To infer the correlation of EGR1-NAB2 expression and HNSCC metastasis, we performed unsupervised hierarchical clustering analysis based on the mRNA expression of these markers, together with EGFR expression as a reference. “